# FORWARD EXPLANATION : WHY CATASTROPHIC FORGETTING OCCURS

## ABSTRACT

The training framework relying on backpropagation and gradient descent has resulted in the creation of opaque models, leading to many problems that we cannot explain. One such problem that has remained inexplicable since the advent of neural networks is catastrophic forgetting. Recently, We have made some intriguing discoveries, which we have integrated into an explanation for neural network training, referred to as Forward Explanation. We first discover that training guides neural networks to produce a particular representation, which we refer to as Interleaved Representation. Additionally, we find that under this representation, neural networks exhibit a series of convergence phenomena, which we term Task Representation Convergence Phenomena. Furthermore, we find that in order to learn this representation, neural networks undergo a specific parameter change during training, which we call Forward-Interleaved Memory Encoding. This unveils some inner workings of how neural networks learn and fundamentally answers why catastrophic forgetting occurs.

## 1 INTRODUCTION

This work stems from our curiosity about the phenomenon of catastrophic forgetting [French (1999),Ramasesh et al. (2020)]. After gaining an understanding of and engaging with modern artificial intelligence, we quickly stumbled upon this intriguing issue. People typically tend to envision artificial intelligence based on common biological knowledge [Kudithipudi et al. (2022)], as often depicted in cultural works, where they possess the ability to continuously learn and analyze, even growing like infants. However, reality promptly informed us that it's not the case. Currently, mainstream artificial intelligence models require pre-input knowledge, and people must prepare all the content that the model needs to learn in advance. Because as the model learns the current material, it forgets what it learned previously. During our research, we were pleased to discover that there are many individuals worldwide who share our concern about this issue. People enthusiastically engage in research to overcome catastrophic forgetting, with a primary focus on the following areas: mitigating catastrophic forgetting [Kirkpatrick et al. (2017),Aljundi et al. (2018)], continual learning [Aljundi (2019),Beaulieu et al. (2020)], incremental learning [Chaudhry et al. (2018),Finn et al. (2017)], lifelong learning [Isele & Cosgun (2018)], and meta-learning [Javed & White (2019)]. While we conducted some research in these areas, it often felt like we were groping in the dark, unable to truly grasp the essence of the problem. We believe that the core challenge of catastrophic forgetting lies in the black-box nature of model training. If we could gain specific insights into how catastrophic forgetting manifests during the training process, perhaps we could develop targeted solutions. This indirectly necessitates an effort to explain the training process of neural networks.

In our experiments, we first observed a highly consistent relationship between the parameters in the final layer of the model and the representations. This led us to suspect whether the two were equivalent, and we experimentally confirmed their equivalence. This directly implies that during the training process, neural networks are effectively training the preceding layers to produce the desired representations. Now, the question arises: what are the characteristics and significance of these representations? Upon further investigation, we found that these representations, in and of themselves, do not possess intrinsic meaning; their primary significance lies in being distinct from other representations. Next, we delved into how these representations are generated. We discovered that neural networks, in their initial state, have convergent representations, which is straightforward since the model's parameters are randomized at this stage. The question then shifts

to understanding what changes occur in the model's parameters during the training process to make the representations of different inputs diverge. To address this question, we introduced the concepts of memory trace and Forward-Interleaved Memory Encoding. The former concept regards input of each layer in the forward propagation as a node, with the first node being the original input and the last node representing the target representation. To achieve divergent representations, one can view it as making this memory trace distinct. The latter concept implies that to achieve this goal, every node on the memory trace must be involved. Differences in the original samples are transmitted layer by layer through the Forward-Interleaved Memory Encoding until reaching the last node.

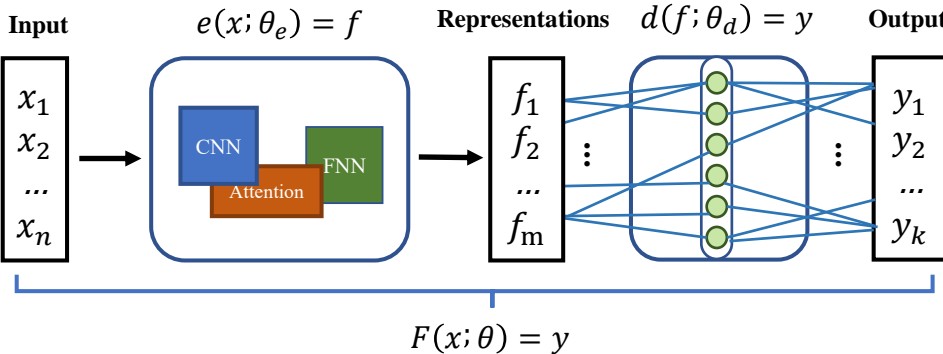

Figure 1: Neural network can be regarded as a combination of an encoder and a decoder.

## 2 FORWARD EXPLANATION

For the training of complex deep neural network models, we generally consider that they are learning some form of mapping from inputs to outputs, denoted as $F(\boldsymbol{x}; \boldsymbol{\theta}) = \boldsymbol{y}$, where $\boldsymbol{\theta}$ represents the model's parameters. However, we do not fully understand how they accomplish this process, which is why they are often referred to as black boxes. To address this issue, we first introduce a definition here. For any model training $F(\boldsymbol{x}; \boldsymbol{\theta}) = \boldsymbol{y}$, we consider it in terms of two sequential components in the spatiotemporal domain. The part closer to the input is referred to as the encoder $e(\boldsymbol{x}; \boldsymbol{\theta}_e) = \boldsymbol{f}$, and the part closer to the output is referred to as the decoder $d(\boldsymbol{f}; \boldsymbol{\theta}_d) = \boldsymbol{y}$. To simplify our discussion, The encoder can take the form of any model, while the decoder is essentially a single fully connected layer network, as illustrated in Figure 1. This does not impact the subsequent discussion regarding forward-interleaved memory encoding; it is merely for experimental convenience.

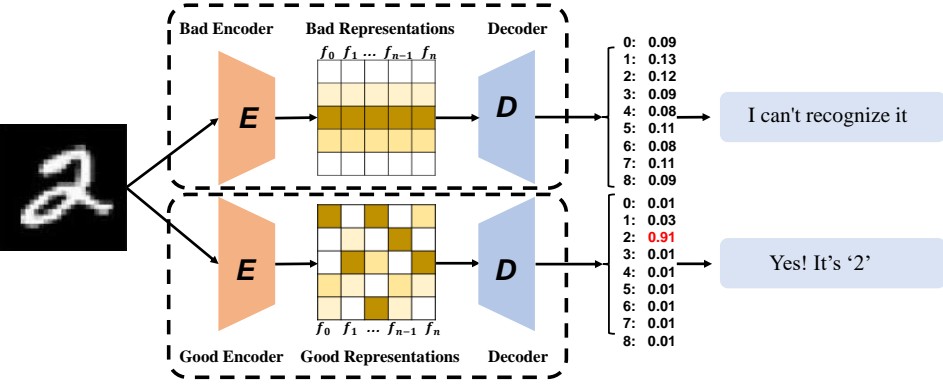

Figure 2: When we say a model performs well on a dataset, it is essentially equivalent to whether the encoder part of the model produces dissimilar task representations for different tasks.

## 2.1 INTERLEAVED REPRESENTATIONS

In order to understand what model training is doing, we must first understand the purpose of model training. Let's consider what representation $\boldsymbol{f}$ the decoder $d(\boldsymbol{f}; \boldsymbol{\theta}_d)$ needs to complete its task, which is the mapping from $\boldsymbol{f}$ to $\boldsymbol{y}$. If representations $\boldsymbol{f}$ obtained from input data of different categories are quite similar when passed through the encoder, then when multiplied by the decoder's weight matrix, their output vectors will also be quite similar. Consequently, the decoder $d(\boldsymbol{f}; \boldsymbol{\theta}_d)$ will be unable to map the same representation $\boldsymbol{f}$ to different $\boldsymbol{y}$ values. Conversely, as long as the differences between representations $\boldsymbol{f}$ are sufficiently large, the decoder should be able to effortlessly map them to different outputs, as illustrated in Figure 2. Based on this intuitive idea, we have devised a series of experiments with the aim of demonstrating the equivalence in terms of their capabilities between the trained model $F(\boldsymbol{x}; \boldsymbol{\theta})$ and the trained encoder $e(\boldsymbol{x}; \boldsymbol{\theta}_e)$.

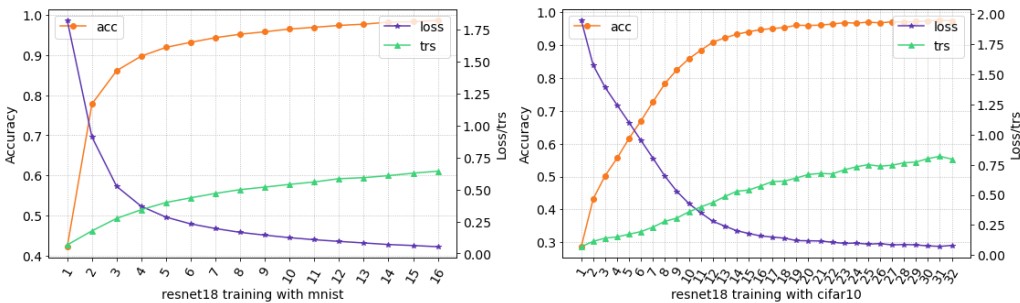

Figure 3: TRS improves as the model performance improves, indicating that training guides the model to possess this characteristic in its output representation.

We define a way to compute the similarity between a set of high-dimensional vectors called task representation similarity (trs). It calculates the variance of the expected representations for each task across each component. This is a relative value, and it does not imply that higher numerical values are indicative of better representations. It solely represents a trend, namely that during a model training process, this value is positively correlated with model performance. This indirectly suggests that the essence of model training is encoding inputs into dissimilar representations, which we refer to as interleaved representations, as shown in Figure 3. We provide a specific definition for it, which means that the expected representations of all independent tasks on each component should ideally be as distinct as possible. In a classification scenario, each individual task corresponds to different categories of samples. We believe that this concept remains consistent in other scenarios as well because decoders cannot do anything meaningful with identical representations.

$$trs(E(\boldsymbol{f}_0, \boldsymbol{f}_1, ..., \boldsymbol{f}_n)) = \frac{1}{m} \sum_{i=0}^{m} var(E(\boldsymbol{f}_0^i), E(\boldsymbol{f}_1^i), ..., E(\boldsymbol{f}_n^i)) \tag{1}$$

$\boldsymbol{f}_i^j$ represents the value of the representation for category $i$ on component $j$, where $E$ represents the expectation, and in the experiments conducted in the paper, the mean is taken.

## 2.2 TASK REPRESENTATION CONVERGENCE PHENOMENA

Here comes an interesting observation: when the representations generated by the encoder are interleaved representations (which is just a tendency and not a fixed form), we have noticed that the decoder exhibits the following two properties. (1) The decoder's performance, whether trained with all representations or with expected representations, remains consistent. (2) The parameters of the decoder converge to the expectation of the representations after being trained, and every component of the weight matrix independently converges to the corresponding representation's expectation. We will demonstrate these two points through the experiments below.

$$training[d(\boldsymbol{f})] \rightarrow training[d(E(\boldsymbol{f}))] \tag{2}$$

$$\boldsymbol{\theta}_d \rightarrow E(\boldsymbol{f}_0, \boldsymbol{f}_1, ..., \boldsymbol{f}_n) \textbf{ and } \boldsymbol{\theta}_d^i \rightarrow E(\boldsymbol{f}_i) \tag{3}$$

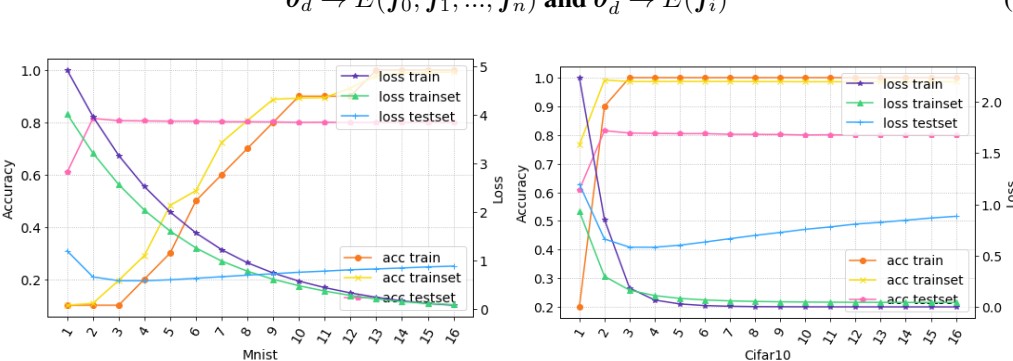

Figure 4: We train $d(\boldsymbol{f}; \boldsymbol{\theta}_d)$ using the compressed $E(\boldsymbol{f})$, and since we only have 10 samples, each training round simply repeats these 10 samples. As a result, the compression or acceleration ratio in training has reached 6000 times.

For Conclusion One, we chose ResNet18 [He et al. (2016),LeCun et al. (1989)] as our experimental model, and MNIST and CIFAR-10 as our experimental datasets. Initially, we trained the model on the target dataset. After pre-trained, we trained a new decoder using the expected representations and subsequently tested this new decoder on the dataset. Taking MNIST as an example, its training dataset contains 60,000 samples, but after compression into expected representations, there are only 10 samples, as illustrated in Figure 4, showcasing the results.

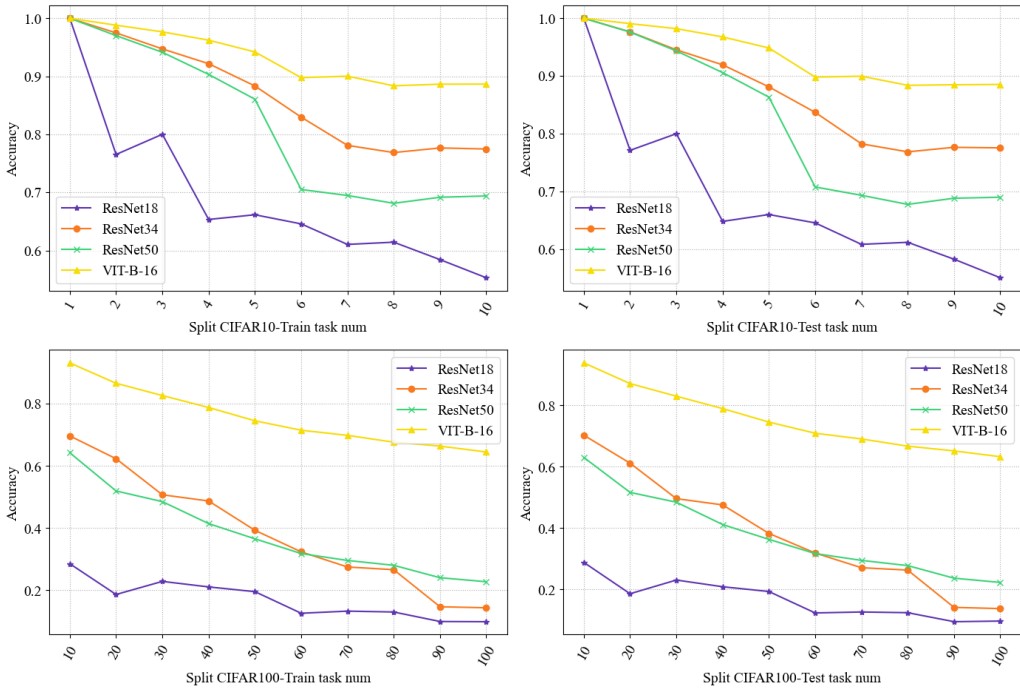

Figure 5: No training has taken place here; instead, for each task $i$, we replace the portion of weight parameters $w$ that belongs to its own category with $E(\boldsymbol{f}_i)$.

Regarding Conclusion Two, we selected ResNet18, ResNet34, ResNet50, and ViT-B-16 [Dosovitskiy et al. (2020),Vaswani et al. (2017)] as our experimental models, and CIFAR-10 and CIFAR-100 as our experimental datasets. Initially, we pretrained these models on the ImageNet1k dataset. Then, we combined the expected representations to create a new decoder and subsequently tested this new decoder on the target dataset. It's worth noting that the pretraining dataset is different from the

target test dataset. This is done to demonstrate the generalizability of Conclusion Two, rather than being tied to a specific dataset. Furthermore, the testing dataset follows a split mode, meaning it is divided into a continuous data stream. For each task $i$, we calculate the expected representation $\boldsymbol{f}_i$ separately, place it in the corresponding position of the decoder's weight parameters, $\boldsymbol{w}_i$, and then test the decoder's performance on all seen tasks, as illustrated in Figure 6. In a sense, this is a particular form of transfer learning that allows the model to perform continuous transfer learning on task-by-task.

The experiments conducted above indicate that when the encoder is capable of producing interleaved representations, the decoder can be acquired with minimal cost. This indirectly suggests that the model training is equivalent to training an encoder. However, we have not provided an explanation for these phenomenon. On one hand, we do not have the time to provide a comprehensive mathematical proof, and on the other hand, this is not the focus of this article. Our primary goal is to use interleaved representations to elucidate how the encoder learns to produce such representations.

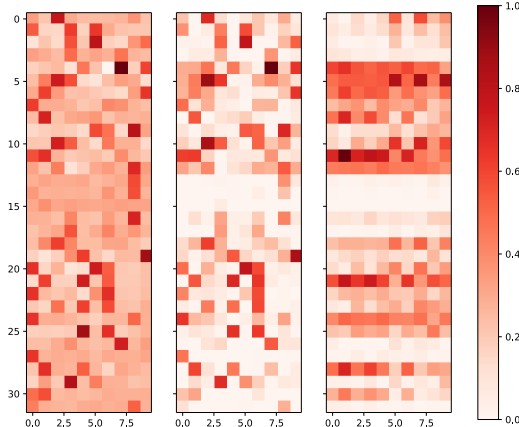

Figure 6: Heatmap1 represents the decoder weight parameters trained on MNIST data, Heatmap2 corresponds to the encoder output $E(\boldsymbol{f}_{mnist})$ after training on MNIST. They exhibit visible similarity. Heatmap3 represents the encoder output $E(\boldsymbol{f}_{fashion})$ on the Fashion dataset after training on the MNIST. At this point, $trs(E(\boldsymbol{f}_{mnist})) = 8.0258$, and $trs(E(\boldsymbol{f}_{fashion})) = 0.6789$.

## 2.3 MEMORY TRACE

In the previous chapter, we transformed the question of how the model is trained to enhance its capabilities into a question of how the encoder is trained to map inputs into interleaved representations. However, before addressing this issue, we need to do some preparatory work. During a single forward propagation pass, a sample $\boldsymbol{x}$ leaves behind a trace $T(\boldsymbol{t}^1, \boldsymbol{t}^2, ..., \boldsymbol{t}^L)$, where $\boldsymbol{t}^l$ represents the input vector to the $l-th$ layer of the network during this forward pass. Such a path is what we refer to as a memory trace. For the backward propagation [Rumelhart et al. (1986)] calculation at any layer, there is

$$\delta_j^l = \sum_K \boldsymbol{w}_{kj}^{l+1} \delta_k^{l+1} \boldsymbol{t}_j'^{l+1} \qquad \frac{\partial C}{\partial \boldsymbol{w}_{jk}^l} = \boldsymbol{t}_k^l \delta_j^l \tag{4}$$

In which $\boldsymbol{w}$ represents the weight parameters of that layer, $C$ is the final loss, $\boldsymbol{t}$ represents the input vector to that layer, and $\delta$ represents the loss of backpropagation. we assume the training structure of the $l-th$ layer network is as follows:

$$\boldsymbol{P} = \begin{array}{c} \boldsymbol{t}_1^l \qquad \cdot \quad \cdot \quad \cdot \qquad \boldsymbol{t}_K^l \\ \left( \begin{array}{cccc} grad(\boldsymbol{w}_{11}^l) & \cdot & \cdot & grad(\boldsymbol{w}_{1K}^l) \\ \cdot & & \cdot & \cdot \\ \cdot & & \cdot & \cdot \\ \cdot & & \cdot & \cdot \\ grad(\boldsymbol{w}_{J1}^l) & \cdot & \cdot & grad(\boldsymbol{w}_{JK}^l) \end{array} \right) \begin{array}{c} \delta_1^l \\ \cdot \\ \cdot \\ \cdot \\ \delta_J^l \end{array} \end{array} \tag{5}$$

$P$ represents the gradient change matrix of the weight parameters $w$ for this layer. As we can see from Equation 4, the changes in $P$ are controlled by $t$ and $\delta$. $\delta$ is determined by the information from the next layer, while $t$ is determined by the information from the previous layer. From the perspective of $t$, all gradient changes necessarily occur in the directions where $t$ is not equal to zero. Furthermore, we can draw the conclusion that in any backward propagation process, all gradient changes of the weights happen in the parameters corresponding to this memory trace $T$.

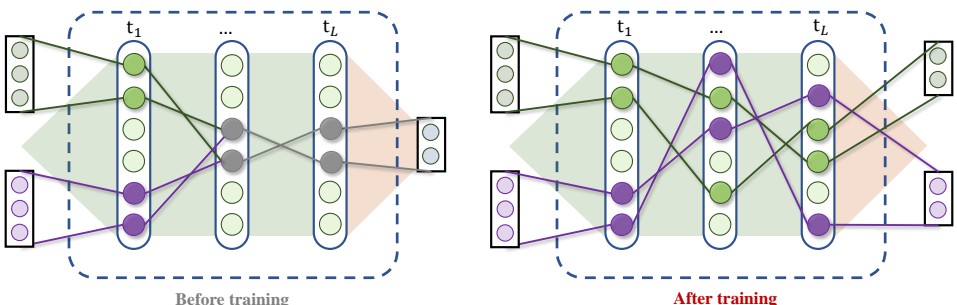

Figure 7: Regardless of whether the neural network's initial parameters are set to constant or random values, the memory trace $T$ for different samples, except for $t_1$, tend to converge. However, after training, their memory trace $T$ diverge. This allows the encoder to ultimately produce interleaved representations.

## 2.4 FORWARD-INTERLEAVED MEMORY ENCODING

First, let's define the training process. Given a batch of samples $X = (\boldsymbol{x}_1, \boldsymbol{x}_2, \boldsymbol{x}_3, ..., \boldsymbol{x}_n)$ of any batch size, they are expected to be decoded into the target $Y = (y_1, y_2, y_3, ..., y_n)$ by the decoder $d(\boldsymbol{f}; \boldsymbol{\theta}_d)$. From the preceding interleaved representation, we can infer that this requires the encoder to encode them into different representations $F = (\boldsymbol{f}_{y_1}, \boldsymbol{f}_{y_2}, \boldsymbol{f}_{y_3}, ..., \boldsymbol{f}_{y_n})$. For any $\boldsymbol{x}_i$ in this batch, its memory trace is denoted as $T_{\boldsymbol{x}_i}$, the memory trace component for the $l-th$ layer is $\boldsymbol{t}^l_{x_i}$, and the loss backpropagation for the $l-th$ layer is $\delta^l_{x_i}$. We can view the encoder $e(\boldsymbol{x}; \boldsymbol{\theta}_e)$ as a series of sub-encoders $e_1(\boldsymbol{t}^1; \boldsymbol{\theta}_{e_1}), e_2(\boldsymbol{t}^2; \boldsymbol{\theta}_{e_2}), ..., e_L(\boldsymbol{t}^L; \boldsymbol{\theta}_{e_L})$, each representing a layer in the memory trace $T$ as a continuous sequence in space and time. Then, for any sub-encoder $e_l(\boldsymbol{t}^l; \boldsymbol{\theta}_{e_l})$, the gradient computation of its weight parameters $\boldsymbol{w}^l_{jk}$ in this round of backward propagation is calculated as

$$\nabla \boldsymbol{w}^l_{jk} = \sum_{i=0}^{n} \boldsymbol{t}^l_{\boldsymbol{x}_i k} \delta^l_{\boldsymbol{x}_i j} \tag{6}$$

Now we can explain what the encoder $e(\boldsymbol{x}; \boldsymbol{\theta}_e)$ is doing during training. In the model initialization phase, all parameters are randomly initialized. At this point, regardless of the input received by the encoder, after undergoing multiple rounds of random parameter adjustment, they will almost all be mapped to similar representations. This implies that, in their memory trace $T$, except for $\boldsymbol{t}^1_{\boldsymbol{x}_i}$, the other components tend to become more consistent the closer they are to the decoder. So, the question of how the encoder generates interleaved representations is transformed into how the neural network, for different samples, adjusts their memory trace $T$ through training to make them diverge rather than converge, as illustrated in Figure 7. The paper assumes default training using the stochastic gradient descent (SGD) algorithm. If other optimization algorithms are used, there may be differences in the formulation process, but it does not affect the conclusion.

For a better discussion, let's assume that in the memory trace $T$ left by different samples after the first forward propagation during training, all $\boldsymbol{t}^2_{\boldsymbol{x}_i}, \boldsymbol{t}^3_{\boldsymbol{x}_i}, ..., \boldsymbol{t}^L_{\boldsymbol{x}_i}$ are the similar, except for $\boldsymbol{t}^1_{\boldsymbol{x}_i}$. For any $e_l(\boldsymbol{t}^l; \boldsymbol{\theta}_{e_l})$, if $\boldsymbol{t}^l_{\boldsymbol{x}_1}, \boldsymbol{t}^l_{\boldsymbol{x}_2}, \boldsymbol{t}^l_{\boldsymbol{x}_3}, ..., \boldsymbol{t}^l_{\boldsymbol{x}_n}$ are all the similar, then at this point, Equation 6 becomes

$$\nabla \boldsymbol{w}^l_j = \boldsymbol{t}^l_{\boldsymbol{x}_i} \sum_{i=0}^{n} \delta^l_{\boldsymbol{x}_i j} \tag{7}$$

which means the same change is made in every output direction. In other words, after this round of training, if two different inputs that entered $e_l(\boldsymbol{t}^l; \boldsymbol{\theta}_{e_l})$ previously resulted in the same output, they

will still be the same now. It merely reinforces the imprint of this memory trace $\boldsymbol{t}^l$ on $\boldsymbol{w}^l$. Even so, they perform an important task - gradient backpropagation. According to our previous definition, during the first backpropagation, layers 2 through $L$ did not produce any meaningful changes.

When the gradient backpropagates to the first layer, a change in the situation occurs. We assume two distinct samples, $x_a$ and $\boldsymbol{x}_b$. According to the previous definition, the memory subpaths $\boldsymbol{t}^2, \boldsymbol{t}^3, ..., \boldsymbol{t}^L$ for $\boldsymbol{x}_a$ and $\boldsymbol{x}_b$ should be similar at this moment. Additionally, to avoid excessive classification, we assume that all variables on the memory trace $T$ are non-negative. Suppose $\boldsymbol{x}_a$ and $\boldsymbol{x}_b$ backpropagate to the first layer with losses $\delta_a^1$ and $\delta_b^1$, respectively. The gradient calculation for the weights $\boldsymbol{w}$ of the first layer $e_1(\boldsymbol{t}^1; \boldsymbol{\theta}_{e_1})$ at each output dimension is determined by equation 6 as follows.

$$\nabla \boldsymbol{w}_{jk}^1 = \boldsymbol{t}_{ak}^1 \delta_{aj}^1 + \boldsymbol{t}_{bk}^1 \delta_{bj}^1 \tag{8}$$

If we consider the positive change in parameter $\boldsymbol{w}$ as reinforcement and the negative change as weakening, then when the directions of $\delta_{aj}^1$ and $\delta_{bj}^1$ are the same, this means that $\boldsymbol{t}_a^1$ and $\boldsymbol{t}_b^1$ both want to reinforce or weaken the component of $\boldsymbol{w}$ in dimension $j$ and make $\boldsymbol{t}_{a_j}^2$ and $\boldsymbol{t}_{b_j}^2$ both high values or low values. When $\delta_{aj}^1$ and $\delta_{bj}^1$ have different directions, it implies that $\boldsymbol{t}_a^1$ and $\boldsymbol{t}_b^1$ engage in a gradient game [Von Neumann & Morgenstern (1947),Nash Jr (1950)] in dimension $j$ of $\boldsymbol{w}$. Furthermore, if the absolute values of $\delta_a^1$ and $\delta_b^1$ do not differ too much, this game can be seen as dominated by the memory trace $\boldsymbol{t}^1$. We further assume that $\delta_{aj}$ is negative, and $\delta_{bj}$ is positive. In this case, when $\boldsymbol{t}_{ak}^1 > \boldsymbol{t}_{bk}^1$, $\boldsymbol{w}_{jk}$ will be reinforced, and when $\boldsymbol{t}_{ak}^1 < \boldsymbol{t}_{bk}^1$, $\boldsymbol{w}_{jk}$ will be weakened. This ultimately results in $\boldsymbol{t}_{aj}^2$ being a high value, while $\boldsymbol{t}_{bj}^2$ is a low value. If such gradient games exist in multiple dimensions, $\boldsymbol{t}_a^2$ and $\boldsymbol{t}_b^2$ will eventually become dissimilar, even if they started as similar. We will provide experiments in Section 3 to demonstrate this phenomenon more intuitively.

From this point, we can derive further. The difference between $\boldsymbol{t}_{x_a}^1$ and $\boldsymbol{t}_{x_b}^1$ causes a change in $\boldsymbol{w}_1$, making $\boldsymbol{t}_{x_a}^2$ and $\boldsymbol{t}_{x_b}^2$ different, which in turn can lead to a change in $\boldsymbol{w}_2$, making $\boldsymbol{t}_{x_a}^3$ and $\boldsymbol{t}_{x_b}^3$ different, and so on. Ultimately, this makes the encoder $e(\boldsymbol{x}; \boldsymbol{\theta}_e)$'s outputs, $\boldsymbol{f}_{x_a}$ and $\boldsymbol{f}_{x_b}$, different. It can be concluded that although backpropagation starts from the last layer, its effective changes to neural network parameters propagate forward from the first layer. Even if we employ any variant of gradient descent optimization algorithms, use any activation function, or adopt any network architecture, two fundamental principles remain unchanged. First, the gradient changes generated for a single sample during one backward propagation step all lie on the memory trace $T$. Second, each sample must undergo forward-interleaved memory encoding with the memory trace of samples from different tasks; otherwise, the neural network cannot distinguish between them.

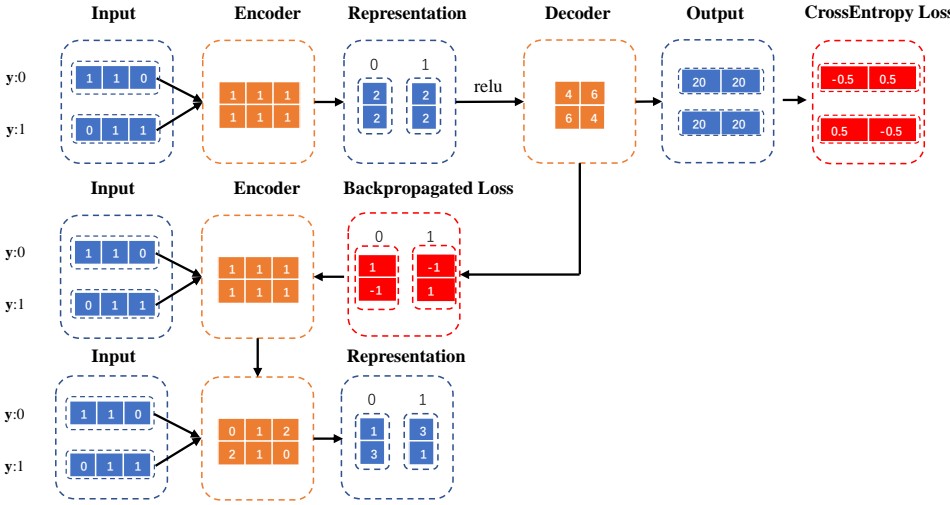

Figure 8: A simple example demonstrating how the encoder distinguishes when the representations are the same.

## 3 WHY CATASTROPHIC FORGETTING OCCURS

It can be fully explained why catastrophic forgetting occurs and how it happens. From the perspective of forward-interleaved memory encoding, we can understand that the essence of neural network training is to differentiate the memory trace of each distinct sample within the network. So, what happens when we train the neural network with the same batch of samples for a long time? The answer is that it continuously reinforces the memory trace for this batch of samples. To be more specific, reinforcement refers to the accumulation of gradients. From the perspective of memory trace, we can observe that all gradient changes are applied to the memory trace, while the parameters in other positions remain unchanged. This leads to two consequences: (1) there may be a significant numerical difference between the parameters on the memory trace and those on the non-memory trace; (2) the historical memory trace do not undergo forward-interleaved memory encoding with the new memory trace. It force historical memory trace to gradually overlap with the new memory trace, ultimately leading to the convergence of the representation.

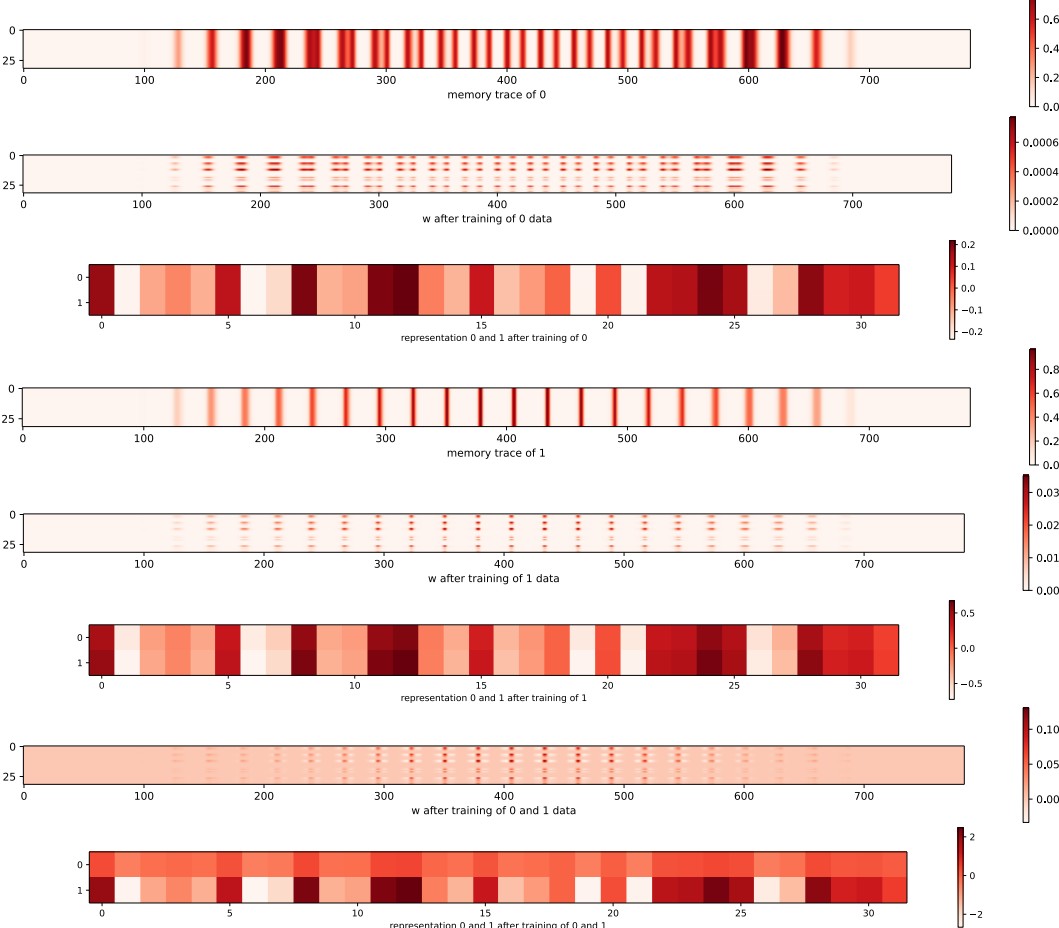

Figure 9: The model will learn the 0-number data subset first, then the 1-number data subset, and finally, the mixed subset of 0s and 1s. There are three noteworthy things here: (1) It's evident that each sample is reinforcing its own memory trace, which is reflected in $w$. (2) Please note that after the model learns the 1-number data subset, the memory trace for the 0-number data on $w$ is not erased. This is because the gradients from the 1-number data cannot interfere with the parameters on the memory trace for the 0-number data. The truth is that there is a significant numerical difference between them, roughly on the order of $10^2$, which causes them to be ignored in the visualization. (3) Only through forward-interleaved memory encoding do their representations become inconsistent.

To clearly observe the imprints of memory trace on the parameters, we initialize all weights to zero, and biases cannot be set to zero simultaneously, as it would prevent loss backpropagation. Therefore, in the experiments, we will observe weight parameters changing only in dimensions with high bias values. For our experiment, we select a subset of the MNIST dataset containing digits 0 and 1 as samples. We flatten the images into one-dimensional vectors and input them into a Multi-layer fully connected network with one hidden layer. The architecture is structured as 784x32x32x10, and the activation function used is ReLU [Glorot et al. (2011)]. We employ the stochastic gradient descent (SGD) optimization algorithm. Our main focus will be on the changes in the first-layer weights and the changes in the representations it outputs.

Now, we have both good news and bad news. The good news is that we now have a clear understanding of how catastrophic forgetting occurs. The bad news is that it is intertwined with the concept of backpropagation. Looking at the results, the root cause of catastrophic forgetting can be traced back to $t_1$, which represents the input to the first layer. Unfortunately, this input is precisely the data we intend to learn. This means that within the training framework of backpropagation, we cannot overcome catastrophic forgetting with a cost equivalent to normal training. This is because we not only need to preserve the memory trace $T$ for each sample but also keep them continuously active to perform forward-interleaved memory encoding on weight parameters. This is almost tantamount to retraining all the data from scratch. From this perspective, it may be necessary to design a new artificial intelligence algorithm to fundamentally address this problem.

## 4 CONCLUSION

We propose an explanatory model for the training of neural networks called Forward Explanation. This theory posits that the essence of neural network training is to acquire a specific kind of representation which we refer to as Interleaved Representation. This representation necessitates that different tasks correspond to representations that differ as much as possible in each dimension, and we have empirically demonstrated this. When given any neural network model $F$ and any dataset $X$, if the representation of $X$ in $F$ is an interleaving representation, then there are task representation convergence phenomenon, as confirmed by our experiments. In summary, our question revolves around how neural networks can be trained to map different inputs into distinct representations.

To address this question, we introduce the concepts of memory trace and Forward-Interleaved Memory Encoding to elucidate this process. It describes how neural networks encode their memories during training. It reveals that the essence of backpropagation is fundamentally a form of forward propagation, where the forward propagation is not about loss but rather the fluctuations in parameters. This ultimately leads to the separation of memory trace for each different sample in the network. The starting point of this trace is the original data, and the endpoint is the output representation. In terms of causality, it is the separation of memory trace that results in distinct final representations. This fundamentally explains the phenomenon of catastrophic forgetting. Moreover, each instance of learning with a task's data accumulates gradients on the parameter trace separately, ultimately forcing the memory trace from historical data to converge towards the memory trace of new data. This fundamentally explains the phenomenon of catastrophic forgetting.

After understanding the reasons behind catastrophic forgetting, can we find a solution to address it? Without altering the framework of backpropagation and gradient descent, we have attempted to tackle this issue from the perspectives of contrastive learning, reinforcement learning, and meta-learning. However, we have found that all these approaches are ultimately constrained by the inability to effectively retain the memory trace from the past. We do not know whether forcefully storing all memory trace would yield any benefits, but it would entail a significant computational and storage overhead, as it would directly correlate with the model's complexity. Anyway, we hope that in the future, genuine artificial intelligence capable of lifelong learning can be realized.

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
