# OpenReview forum: "Forward Explanation : Why Catastrophic Forgetting Occurs"
_ICLR.cc/2024/Conference — Submitted to ICLR 2024_

### Official Review · Reviewer_Jw5d · 2023-10-27

**Soundness:** 1 poor
**Presentation:** 1 poor
**Contribution:** 1 poor
**Rating:** 1
**Confidence:** 5

**Summary:**

In order to uncover the root causes of catastrophic forgetting in Continual Learning, this study analyzes phenomena that arise during single task training. This analysis reveals three key phenomena (Interleaved Representation Production, Task Representation Convergence Phenomena, and Forward-Interleaved Memory Encoding) inherent in the training process of neural networks.

**Strengths:**

None

**Weaknesses:**

W1. The paper presents unsubstantiated claims and contents. We provide some examples of unsubstantiated claims:

(Section 2.2) “This indirectly suggests that the model training is equivalent to training an encoder. However, we have not provided an explanation for these phenomenon. On one hand, we do not have the time to provide a comprehensive mathematical proof,”
(Section 2.4) “In the model initialization phase, all parameters are randomly initialized. At this point, regardless of the input received by the encoder, after undergoing multiple rounds of random parameter adjustment, they will almost all be mapped to similar representations.”
(Section 2.4) “The paper assumes default training using the stochastic gradient descent (SGD) algorithm. If other optimization algorithms are used, there may be differences in the formulation processes, but it does not affect the conclusion.”


W2. The phenomena claimed to be elucidated in the paper are already well-known facts, rendering them trivial. We provide some examples:

- The observation in Section 2.1, where distinct representations are formed based on classes, is a well-established fact.
- The practice of employing the expectation of a representation as a decoder in Class Incremental Learning is a widely recognized approach. For more details, you can refer to the following paper. [Rebuffi, S. A., Kolesnikov, A., Sperl, G., & Lampert, C. H. (2017). icarl: Incremental classifier and representation learning. In Proceedings of the IEEE conference on Computer Vision and Pattern Recognition (pp. 2001-2010)].


W3. The explanation in the paper is difficult to comprehend.

- The notations are confusing.
- The provided explanations for the experimental results are insufficient.
- Descriptions for the axes on the figures are absent.

**Questions:**

Please address the concerns raised in the 'Weaknesses' section.

---

> ### Author Response · Authors · 2023-11-11
>
> Response to Weak1:
> (1) There may be a bias in the expression. We simply aim to convey that training is primarily to enable the encoder to acquire the ability to differentiate inputs.
> (2) This part does lack additional experiments, but it is indeed the case.
> (3) If other optimization algorithms are used here, it is true that it won't be as straightforward as in Section 2.4. However, the change will be more complex, but it still occurs along the memory trace. If forgetting occurs, similar representations will be generated, and if normal training takes place, dissimilar representations will be produced.
>
> Response to Weak2: Thank you for your information.
>
> Response to Weak3: Thank you for your feedback.

---

### Official Review · Reviewer_T5LT · 2023-10-29

**Soundness:** 2 fair
**Presentation:** 2 fair
**Contribution:** 1 poor
**Rating:** 1
**Confidence:** 4

**Summary:**

The authors present an empirical investigation on catastrophic forgetting, mostly by tracking changes in representations learnt by neural networks during training. In particular, they look at both the impact of last-hidden-layer representations ('encoder' in the paper), and at the changes in activation in all layers of the network during training.
For example, they observe that the last-hidden-layer representations become class-discriminative (which is however trivially implied, since the output layer acts as a linear classifier, and accuracy improves during training).

**Strengths:**

The authors thoroughly look at the problem of catastrophic forgetting and in general of neural network training in a thorough fashion.

The article is written in an engaging way (but see 'Weaknesses' below).

**Weaknesses:**

Overall, it seems that none of the results in the paper are particularly original, and it rather looks like that the authors are not aware of prior work in the field.

This is reflected by some mis-use of terminology (e.g., encoder/decoder) and by strong statements (e.g., "One such problem that has remained inexplicable since the advent of neural networks is catastrophic forgetting"; "However, we do not fully understand how they accomplish this process, which is why they are often referred to as black boxes.").

Likewise, language is informal and imprecise (e.g., introduction, and then at many places in the text like "To address this issue, we first introduce a definition here"; "Now, we have both good news and bad news. The good news is that we now have a clear understanding of how catastrophic forgetting occurs. The bad news is that it is intertwined with the concept of backpropagation".
The way the article is written would be suitable for a blog post, but not for a formal scientific article.

General remarks:
    - The split of any neural network in encoder/decoder parts seems mostly arbitrary and it does not add much to the paper.
    - It seems that most of the results presented have been known in the field for a long time?

    - Typos and general grammar issues, e.g., "After pre-trained, we trained a new decoder using", etc.
    - Minor: fig. 4 use of accuracy may be a bit misleading, since only 10 samples are used (it is also not clear whether the full test set is used, or also only 10 compressed representations like for the training set). In this context, `accuracy' can only take 11 possible values.

**Questions:**

Is the memory trace just the collection of all layer activations?

---

> ### Author Response · Authors · 2023-11-11
>
> Response to Weaks: Thank you for your feedback.
>
> Response to the question: Yes , it comes in two forms: one as a input vector and the other as traces embedded within the parameters.

---

### Official Review · Reviewer_tGEX · 2023-10-29

**Soundness:** 1 poor
**Presentation:** 1 poor
**Contribution:** 1 poor
**Rating:** 3
**Confidence:** 5

**Summary:**

The paper carries on an analysis on the catastrophic forgetting issue in neural networks. The authors claim that the provided explanation, referred to as **Forward explanation**,  sheds light into the forgetting problem.

**Strengths:**

Investigating the catastrophic  forgetting issue is very relevant to the community.

**Weaknesses:**

Despite the relevance of the topic, the paper contributions are very poor. The paper lacks a proper structure and the experimental  investigation is not sufficient for a conference like ICLR. There are many bold claims that are not supported by sufficient experiments (the conducted experiments are not well described and are extremely simplicistic).
Even the main intuitions are misleading (e.g. dissimilar representeation are referred to as **interleavead**, which stands for the opposite) or already known from the dawn of the field [1,2]

The authors only give intuitions about the main concepts without any experimental validation: in the the Task Representation Convergence section, the authors claim that  *"the parameters of the decoder converge to the expectation of the representation"* . What does this mean?

Similarly, the procedure of training the decoder  using the *expected representation*  is only mentioned without a proper definition.
Overall, the paper lacks a sufficient structure and carries on a story-telling rather than a proper and rigorous theoretical/experimental analysis.

[1]  McCloskey, Michael, and Neal J. Cohen. "Catastrophic interference in connectionist networks: The sequential learning problem." Psychology of learning and motivation. Vol. 24. Academic Press, 1989. 109-165.

[2] French, Robert M. "Catastrophic forgetting in connectionist networks." Trends in cognitive sciences 3.4 (1999): 128-135.

**Questions:**

1) The authors only give intuitions about the main concepts without any experimental validation: for instance, in the the Task Representation Convergence section, the authors claim that  *"the parameters of the decoder converge to the expectation of the representation"* . What does this mean? All over the paper text there are similar issues.

2) The procedure of training the decoder  using the *expected representation*  is only mentioned without a proper definition. The authors should clarify all the definition and better formalize them.

3) When tackling *Conclusion 2* (bottom part of page 4), the authors claim that *"the pretraining dataset is different from the target test dataset"*. There are no additional details on what the authors are describing.

---

> ### Author Response · Authors · 2023-11-11
>
> Response to Weaks:
>
> The term "interleaved representations" is inspired by Section 2.4. From the content of Section 2.4, we can understand that the numerical distribution of representations between different tasks will exhibit an interleaved pattern of highs and lows. The implementation details of the expectation of representation are mentioned in the final part of Section 2.1.
>
> Part of this is indeed missing in Section 2.2, as reflected in Figure 6. If we use the same dataset for testing in the experiments of Equation 3 as in the pre-training dataset, the accuracy curve will stay around 100%. This is why we claim that "the parameters of the decoder converge to the expectation of the representation"
>
> Response to the question: Thank you for your feedback.

---

### Official Review · Reviewer_xF7Q · 2023-11-02

**Soundness:** 1 poor
**Presentation:** 1 poor
**Contribution:** 2 fair
**Rating:** 1
**Confidence:** 5

**Summary:**

In this work, the authors try to understand the working on NNs to understand better why NNs catastrophically forget. They conduct experiments on mnist and cifar-10 to show how representation changes over time on various tasks.

**Strengths:**

1.	Tackles important problem of understanding forgetting in NNs.

**Weaknesses:**

1. Weak Related work section
2. Novelty is limited
3. Writing can be improved-- Text is copied from some thesis, as it talks about chapter, conclusion 1, 2 etc, and none of those are mentioned in the main paper.
4. The experimental section is weak.
5. No validation set in the code shows that the hyper-parameter is optimized on the test set, which would create a biased outcome
6. The test set is shuffled
testloader = data.DataLoader(DATASET(train=False,task=task),batch_size=64,shuffle=True,num_workers=4), why would you shuffle the test set?

**Questions:**

There is a large body of theoretical work neglected in this work. As the issue of catastrophic forgetting is well studied theoretically [1-8], even the issues of neural cross-talk across task is studied empirically [9-11]. Hence, I am not sure about the novelty of this work, as almost all concepts covered in this work have already been proven in prior works so what is the novelty?

Additionally, it looks like the current manuscript is borrowed from a thesis, as section 2.3 says, “In the previous chapter”—What is the chapter here?

What are Conclusion One and Conclusion 2? Why are they not mentioned in the text? On page 3, the author talks about conclusion one? However it is not mentioned anywhere in the manuscript what the conclusion means.

Experiment details are missing; nothing is mentioned to reproduce the results, no ablation study, and no discussion on hyper-parameter optimization.

Forgetting in these systems can also be understood by forgetting ratio, backward transfer, and forward transfer (for instance, look at gradient episodic memory paper).

The current draft is not ready to get published in conferences such as ICLR; it lacks structure, the related section is weak, the experimental section is weak.

There is a plethora of work that tries to explain the working of memory, representation and internal of NNs, I would advise looking into literature.


1.	Doan, T., Bennani, M.A., Mazoure, B., Rabusseau, G. and Alquier, P., 2021, March. A theoretical analysis of catastrophic forgetting through the ntk overlap matrix. In International Conference on Artificial Intelligence and Statistics (pp. 1072-1080). PMLR.

2.	Raghavan, K. and Balaprakash, P., 2021. Formalizing the generalization-forgetting trade-off in continual learning. Advances in Neural Information Processing Systems, 34, pp.17284-17297.

3.	Evron, I., Moroshko, E., Ward, R., Srebro, N. and Soudry, D., 2022, June. How catastrophic can catastrophic forgetting be in linear regression?. In Conference on Learning Theory (pp. 4028-4079). PMLR.

4.	Mirzadeh, S.I., Chaudhry, A., Yin, D., Hu, H., Pascanu, R., Gorur, D. and Farajtabar, M., 2022, June. Wide neural networks forget less catastrophically. In International Conference on Machine Learning (pp. 15699-15717). PMLR.

5.	Braun, L., Dominé, C., Fitzgerald, J. and Saxe, A., 2022. Exact learning dynamics of deep linear networks with prior knowledge. Advances in Neural Information Processing Systems, 35, pp.6615-6629.

6.	Lin, S., Ju, P., Liang, Y. and Shroff, N., 2023. Theory on Forgetting and Generalization of Continual Learning. arXiv preprint arXiv:2302.05836.

7.	Andle, J. and Yasaei Sekeh, S., 2022, October. Theoretical understanding of the information flow on continual learning performance. In European Conference on Computer Vision (pp. 86-101). Cham: Springer Nature Switzerland.

8.	Heckel, R., 2022, May. Provable continual learning via sketched Jacobian approximations. In International Conference on Artificial Intelligence and Statistics (pp. 10448-10470). PMLR.

9.	Ororbia, A., Mali, A., Giles, C.L. and Kifer, D., 2022. Lifelong neural predictive coding: Learning cumulatively online without forgetting. Advances in Neural Information Processing Systems, 35, pp.5867-5881.

10.	Serra, J., Suris, D., Miron, M. and Karatzoglou, A., 2018, July. Overcoming catastrophic forgetting with hard attention to the task. In International conference on machine learning (pp. 4548-4557). PMLR.

11.	Driscoll, L.N., Duncker, L. and Harvey, C.D., 2022. Representational drift: Emerging theories for continual learning and experimental future directions. Current Opinion in Neurobiology, 76, p.102609.

---

> ### Author Response · Authors · 2023-11-11
>
> Response to Weak6: The dataset content is controlled by the parameter [task] here. For instance, [0] indicates only dataset 0, while [0,1,2] includes datasets 0, 1, and 2. Therefore, the use of SHUFFLE has no impact. When we specifically desire a certain dataset, it is controlled through [task].
>
> Response to Question 1: We have provided an explanation of the relationship between representations, parameters, and catastrophic forgetting. Then we try to reproduce how catastrophic forgetting manifests in parameter changes during the training process. This is distinct from [1-11]. The contribution lies in a negative assessment, asserting that without altering the training method, it would be impossible to overcome  catastrophic forgetting at the cost of retraining, as referenced in Section 3, paragraph 3. Simultaneously, it suggests directions for improvement. To effectively overcome catastrophic forgetting at a reasonable cost, one must utilize the manifestation of historical memory trace in network parameters, aiming to achieve effects similar to those described in Section 2.4.
>
> Response to Question 2: This is a writing issue and specifically refers to Sections 2.1 and 2.2.
>
> Response to Question 3: This is a writing issue and actually refers to equation 2 and 3.
>
> Response to Question 4: Regarding the experiments in Section 2.2, we have provided the source code. For Conclusion One, parameters can be freely adjusted, while for Conclusion Two, there is no training process involved as it is computed directly. As for the experiments in Section 3, we utilized a third-party visualization library, so the source code is not included.
>
> Responses to Questions 5-7: We aim to establish a simpler logical chain to illustrate how catastrophic forgetting specifically occurs in engineering.

---

### Author Response · Authors · 2023-11-11

Overall, this article arrives at a negative conclusion because we have only identified the root cause of the problem but currently have not devised effective solutions. We aim to share this discovery, informing those researching this relevant topic.

The reason why datasets in training must be treated as a whole is that their memory trace[2.3] need to be collectively activated across all network parameters [2.4] to generate effective representations [2.1]. This implies that the source of catastrophic forgetting stems from historical memory trace not participating in new training [2.4]. The simplest way to access historical memory trace is by involving historical data in the training again. This leads to a crucial conclusion: the cost of any method avoiding [2.4], considering the overall effectiveness, may be greater than normal training. This suggests that many might be engaged in activities without meaningful outcomes. Conversely, if we can effectively utilize historical memory trace, in whatever form, whether compressed and stored externally or as traces embedded within the parameters, it becomes a practical path toward lifelong learning.

---

### Meta-Review · Area_Chair_c3PS · 2023-12-13

**Metareview:**

This paper aims to come up with a theory or explanation for why catastrophic forgetting happens while training neural networks in a continual learning setting.

All the reviewers agree that the paper is not ready for publication. Reviewers highlight that the paper ignores the rich literature on catastrophic forgetting that already exists. The authors must spend time positioning their findings with respect to this already existing literature. The experiments are also weak and the paper has a lot of bold claims which needs to be backed up by thorough experiments. Authors' rebuttal was not convincing.

I recommend a rejection and also encourage the authors to work on the feedback by the reviewers to have a stronger next submission.

**Justification For Why Not Higher Score:**

There are several issues with the paper: lack of literature review, no strong experiments, and claims that cannot be supported by the experiments.

**Justification For Why Not Lower Score:**

N/A

---

### Decision · Program_Chairs · 2024-01-16

Reject